# European IoT Use in Homes: Opportunity or Threat to Households?

**DOI:** 10.3390/ijerph192114343

**Published:** 2022-11-02

**Authors:** Idiano D’Adamo, Assunta Di Vaio, Alessandro Formiconi, Antonio Soldano

**Affiliations:** 1Department of Computer, Control and Management Engineering, Sapienza University of Rome, Via Ariosto 25, 00185 Rome, Italy; 2Department of Law, University of Naples Parthenope, Via G. Parisi 13, 80132 Naples, Italy; 3Sapienza University of Rome, 00185 Rome, Italy

**Keywords:** digital society, Europe, households, Internet of Things, multi-criteria decision analysis

## Abstract

The residential sector is characterized by new digital challenges. The Internet of Things (IoT) is a key-driver of innovation and operations management. This study aims to measure and assess IoT devices at the level of individuals, which are households, in European countries. For this scope, through the multi-criteria decision analysis (MCDA), we analyse data from Eurostat providing a mix of indicators allowing information to be aggregated at the level of individual Europeans and disaggregated by age group. The results highlight that only four countries (Netherlands, Denmark, Sweden and Malta) are classified as a high cluster in the examined scenarios. The 16–24 age group is the most involved in the uses of IoT devices, but the previous three northern European countries also show very high values for the 35–44 age group. IoT devices serve as a springboard for achieving a powerful propulsion toward technological innovation in the new business models, identifying opportunities and being a way to make many routine tasks more agile. Training programs and awareness campaigns are policy suggestions for the development of IoT devices favouring a cultural change on their use. However, there is an emerging need for studies that monitor environmental health impacts to prevent possible threats.

## 1. Introduction

Our current “digital age” is defined by information and communication technology (ICT), which includes a wide range of connected soft and hardware processes. The way people interact with one another and their surroundings has changed [1,2]. The pandemic period has placed humans into a scenario in which in a short time everything can change and even personal freedoms can be curtailed for the common welfare [3]. However, different thoughts and points of view open up on this issue. The pandemic period could drive the spread of artificial intelligence and robotics [4]. The antifragility of enterprises can propel them to survive and thrive, even after pandemic periods, when there is a link between research and innovation [5]. Innovation is able to generate benefits [6], capturing the green transition by means of operations management [7], the mix of gender diversity [8], the role of industrial districts [9], the impact of Big Data (BD) and service innovation [10].

Digitalization has the unimaginable potential—both favourably and negatively—to contribute to the sustainability of the human and planetary systems, or at the very least, to lessen their negative effects. Digitalization and new technologies promote a more efficient use of resources, improving business competitiveness, as well as the relationship between economic growth and sustainable yield [11]. The world of ecosystems is complex, composed of multiple alchemies and relationships, in which cascading effects can be generated [12], and for some scholars, ICT and BD can support sustainability, and the decision making processes of organisations [13,14]. Specifically, BD allows the enterprise to manage huge amounts of data that are processed through the most modern analysis systems, favouring the interweaving of the information contained in the database, creating the conditions for the enterprise’s competitive advantage [15]. Technological innovation is configured as a governance issue that influences the business model by inviting the enterprise to develop strategies capable of meeting the needs of the context [16]. The financial and innovation performances of enterprises are positively impacted by organizational agility [17]. The use of BD allows an open management of business processes which, through the involvement of interested parties [18], also favours the achievement of sustainability objectives by developing corporate social responsibility [19,20].

It determines the need of a transdisciplinary approach [21]. Likewise, to decide about the technological options to be adopted, business organisations can cross-evaluate the indicators and the characteristics of sustainable development of each product and operating process to facilitate informed decisions aimed at achieving a sustainable performance [22]. Digital technologies, in the form of e-health services, robotics, or emission reduction solutions could help individuals, organizations, and nations achieve sustainability goals [23]. However, it should be pointed out that in the absence of a change in attitude, less selfishness, and a sustainable hand approach, the goals are difficult to achieve [24]. The main concern is that both sustainability and digitization can bring changes, improvements but if they do not manifest themselves in the same way in different countries, they create differences.

The Internet of Things (IoT) is considered an enabler for enterprise digitization strategies [25,26]. Likewise, in the challenges from the environmental change for the business organisations [27], the adoption of cloud computing, BD, as well as ICT, where the main pillar is IoT [15,26] is a “must” for survival. Hence, the IoT is viewed as a component of the future Internet and will include billions of intelligent, talkative “things”. A heterogeneously connected network of devices will make up the Internet of the future, significantly extending the boundaries of the globe with both real and virtual elements [28]. The literature analysing the IoT highlights that it is an increasingly relevant topic and examines corporate and manufacturing sectors, but also what pertains to the home, health care, and knowledge management [29,30]. Its benefits are many as it enables redesigning business processes [31], supporting appropriate business models [32], and providing value to value chain transformation [33]. The technology theme has led enterprises to redefine their boundaries with a greater focus on the external environment not composed only of the customer [24]. New business model approaches have been required for this goal [34], and the contribution made by technology has been relevant [35]. However, issues related to environmental health need to be evaluated to identify a proper mix between the needs of a civil society and those of ecosystems [36]. The approach encompassing sustainable transition changes, are also reflected in digital ones, relative to the residential [37] or industrial applications [38], but also to cross-cutting [39].

Smart grid and smart house applications benefit greatly from the IoT, compared to traditional communication technologies, however these applications are still very uncommon [40]. It plays a key-role in a smart city [41]. The IoT market is characterized by strong growth and, according to the analysis reported by Mordor Intelligence, this market has a strong fragmentation of competitors. This allows even small and medium-sized enterprises to be able to enter this sector. In addition, the Compound annual growth rate (CAGR) is 10% with data for the Asian, North and South American, and African basins. Thus, an emerging theme is to analyse the use of IoT devices in homes in Europe as well [42]. The adoption of the IoT would seem to be poorly adopted, in terms of the data collection of a socio-environmental nature (i.e., PM10 in the workplace; CO_2_ emissions into the atmosphere, etc.) and there is a lack of consolidated practices for measuring performance in this field [15]. Thus, the relationship between the process performance measurement systems and the IoT requires more attention [43]. Some approaches consider its link to sustainability goals [44], others to the supply chain [45]. However, analysis of the literature on the use of indicators to evaluate the performance of the IoT applications in households reveals a gap. In order to solve it, we highlight how the literature proposes the use of multi-criteria decision analysis (MCDA) to compare the performance of European countries [46] and how it is necessary to use a mix of methodologies when the issue to be solved concerns multiple perspectives [47]. The aim of this study is to assess the use of IoT devices in households in Europe, starting from Eurostat data and using several indicators that are able to provide multiple information: the ranking of the IoT use, the distribution by age group, and identification of clusters.

The study is organized as follows. Section 2 provides materials and methods. Section 3 describes the results. Section 4 includes practical and theoretical insights, and the conclusion.

## 2. Materials and Methods

Multi-criteria analysis is widespread in the literature [48]. In fact, it allows alternatives to be compared with each other in order to provide additional information to decision makers. Its approach also involves comparing spatial realities with each other to assess whether or not any clusters exist [49]. The main advantage of this method is that it can aggregate a very large amount of information, while it does not evaluate existing interactions among several variables.

The data considered were collected from the Eurostat portal, which provides access to a wide collection of data, characterized by a high reliability [47]. This tool is useful because it harmonizes the statistical methods across member states. The following flow was considered: Science, technology, digital society → Digital economy and society → ICT usage in households and by individuals → Internet of Things.

We have considered all the items available in Eurostat and it is appropriate to use a single database in order to maintain the homogeneity of the data considered. Subsequent analyses could expand this dataset but paying as much attention to not repurposing redundant data or data collected by statistically different methods that could compromise the available dataset. This approach is justified by the literature [47].

The dataset presents data updated to 2020, regarding the use of IoT devices as a percentage of the entire population of each European Union country. Within the dataset, there is a large number of sheets, each referring to the type of individual analysed, broken down by age, gender, occupation, type of profession, level of schooling, income brackets, living area and citizenship. The countries present are 26 of the EU 27, where France is excluded because data for that country are not reported.

### 2.1. Category and Criteria Identification

The chosen database was composed of datasheets divided by various criteria, as anticipated earlier. For this project, it had been chosen to focus on the analysis of the IoT device market for various age groups, resulting in a categorization, based on contiguous and distinct age groups. In addition, it had been decided not to use datasheets related to categories for which there was not a consistent and meaningful amount of data, thus eliminating age groups with an excessive amount of unavailable data. These bands were the population under 15 years of age and the population over 75 years of age. The following bands were selected:All individuals.Individuals, 16 to 24 years old.Individuals, 25 to 34 years old.Individuals, 35 to 44 years old.Individuals, 45 to 54 years old.Individuals, 55 to 64 years old.Individuals, 65 to 74 years old.

Furthermore, among the 26 countries covered by the analysis, issues emerged during the data extrapolation. For Ireland, the missing data covered only the 16–24 age group, while for Belgium, the data present referred only to certain criteria (about half) for different age groups. For this reason, Belgium was also not included in the analysis.

The criteria identified were related to various types of IoT devices. The criteria selected were twelve:Individuals used internet-connected thermostat, utility meters, lights, plug-ins or other internet-connected solutions for energy management for their home.Individuals used internet-connected home alarm system, smoke detectors, security cameras, door locks or other internet-connected security/safety solutions for their home.Individuals used internet-connected home appliances, such as robot vacuums, fridges, ovens, coffee machines.Individuals used a virtual assistant in the form of a smart speaker or of an app.Individuals used the internet on an internet-connected TV in their home for private purposes.Individuals used the internet on an internet-connected game console.Individuals used the internet on an internet-connected home audio system, smart speakers.Individuals used the internet on an internet-connected TV, game console, home audio system, smart speakers.Individuals used a smart watch, a fitness band, connected goggles or headsets, safety-trackers, internet-connected accessories, internet-connected clothes or shoes.Individuals used internet-connected devices for monitoring blood pressure, sugar level, body weight or other internet-connected devices for health and medical care.Individuals used toys connected to the internet, such as robot toys (including educational) or dolls.Individuals used a car with a built-in wireless internet connection.

The proposed criteria included another (individuals have not used any of the internet-connected devices or systems) that was discarded because it proposed information opposite to the other criteria.

### 2.2. A Mix of Indicators

The methodological approach used in this study is not only based on the MCDA. In fact, simple indicators that can provide different assessments are also proposed to enable us to achieve the research objective of this study. To have a comprehensive view, we can certainly make use of the summary data, but different tools are needed.

The research question (RQ) that this work aims to investigate is the use of IoT devices in European households. In order to achieve this goal, a mix of (ST) Statistical Tools is used to ensure accuracy and reliability in the results. The percentage data are comparable to each other, and therefore the first indicator (ST1) to be used is a simple arithmetic mean.
(1)$${\mathrm{AVG}}_{i,j}=\sum_{c=1}^{N_{C}}I$$
in which, AVG = average; V = value; i = country; j = age group; c = criteria and N_C_ = number of criteria. This enables the criteria to be compared with each other without altering the numerical comparison value. This is an allowed option, as all values express a percentage value ranging from 0 to 100.

The second step (ST2) in the analysis requires understanding how well the value performs in a specific age group. A new RAVG indicator is introduced that represents the ratio of a country’s average in a specific age group to the highest value that the same country obtains in all age groups. For example, Italy (IT), for different age groups has the following values:$${\mathrm{AVG}}_{\mathrm{IT},16–24}=19,{\mathrm{AVG}}_{\mathrm{IT},25–34}=16,{\mathrm{AVG}}_{\mathrm{IT},35–44}=15,{\mathrm{AVG}}_{\mathrm{IT},45–54}=11,{\mathrm{AVG}}_{\mathrm{IT},55–64}=8{\;\mathrm{and}\;\mathrm{AVG}}_{\mathrm{IT},65–74}=4.$$

The maximum value is ${\mathrm{AVG}}_{\mathrm{IT},16–24}$ and consequently ${\mathrm{RAVG}}_{\mathrm{IT},16–24}=1.$
(2)$${\mathrm{RAVG}}_{i,j}={\mathrm{AVG}}_{i,j}/\max\left({\mathrm{AVG}}_{i}\right)$$

However, to have a method that is also applicable to criteria that have different units, the MCDA method can be applied. This requires calculating a row vector (RV) and a column vector (CV) identifying a summary value. This indicator is the normalized result related to the impact of the different criteria (NAVG_i,j_). This represents the third phase of analysis (ST3). The row vector has a dimension (1, N_C_) measuring the value of the criteria, while the column vector (N_C_, 1) quantifies the weight of the criteria. Compared with the percentage approach used previously, a normalization approach is proposed at this stage. This method involves assigning the value 1 to the highest criterion value and 0 to the lowest criterion value. All intermediate values by the linear interpolation are calculated with reference to the maximum and minimum values. The row vector is then populated by the normalized values, which are twelve in number. One value corresponding to each criteria. As for the column vector, initially using an equal-weight approach, the single component is 0.083 (i.e., one divided by twelve).
(3)$${\mathrm{NAVG}}_{i,j}=\mathrm{RV}\times \mathrm{CV}$$

Accordingly, we can proceed to the fourth proposed step (ST4) to investigate the impact of age groups, in terms of individual performance with respect to the country with the highest use of IoT devices. The proposed new indicator ∆Gap_i,j_ is calculated as the difference between the maximum value obtained in a specific age group for all countries and the average obtained in the same age group for a country.
(4)$$∆{\mathrm{Gap}}_{i,j}=\max\left({\mathrm{AVG}}_{j}\right)-{\mathrm{AVG}}_{i,j}$$

For example, considering the age group 35–44 in the Netherlands (NL) has the highest value (${\mathrm{AVG}}_{\mathrm{NL},35–44}=31$) and Malta (MT) has a lower performance (${\mathrm{AVG}}_{\mathrm{MT},35–44}=29$). Consequently, the ∆Gap_i,j_ for the two countries will evidently be zero for the former ($∆{\mathrm{Gap}}_{\mathrm{NL},35–44}$) and two for the latter ($∆{\mathrm{Gap}}_{\mathrm{MT},35–44}$).

Finally, the last analysis aims to assess whether it might be more correct not to use a weighted average approach (ST5). Weights can be assigned through experts or by resorting to objective data (e.g., enterprise or laboratory). In this study, we will make use of the data identifiable from the Google search engine in the alternative scenario. In fact, the literature analysis [47] suggests this approach is suitable, particularly for the digital components, in which the component to be considered is entered as keywords. The Equation (5) is the same as Equation (3), only the column vector changes in which the individual criteria will weigh differently (CV_W_). Consequently, the indicator NWAVG_i,j_ will provide a different evaluation than NAVG_i,j_.
(5)$${\mathrm{NWAVG}}_{i,j}=\mathrm{RV}\times {\mathrm{CV}}_{W}$$

## 3. Results

The results present a baseline scenario in which the ranking of European countries is proposed by considering the overall data (Section 3.1) and then proceeding to evaluate the distribution of the data by age group (Section 3.2). This leads to proposing a cluster of European countries by the MCDA, considering both the overall value (Section 3.3) and the specific value by age group (Section 3.4). Finally, Section 3.5 presents the results of the alternative scenario, based on the MCDA.

### 3.1. Europe-Wide Analysis for All Age Groups

In the first analysis (ST1), in which the weights of the criteria are the same (baseline scenario), an analysis is first conducted at the overall level and then at the level of the individual age groups. The mean value concerns all types of devices and is conducted for 25 countries—Figure 1. Separate maps for each age group are shown in Figure 2. The indicator used in this analysis was the arithmetic mean AVG_i,j_.

The results found an uneven distribution of the IoT device usage rates. In particular, it can be seen that:regarding the Northern European countries, Denmark (second position) and Sweden (third position) have very high performances, and Finland, Ireland and Estonia also show significant values. Latvia and especially Lithuania have less significant performances.as for the Central European countries, the Netherlands (occupying the first position) and Luxembourg show very positive performances with Germany and Austria. Instead, Hungary and Poland have a less significant value, but Romania occupies the second-to-last position. Slovakia and the Czech Republic have an average performance.as for the southern European countries, Spain and Malta, but also Portugal and Slovenia have a good performance, unlike Italy and Greece (second to last place). Croatia has an average performance.

This division of Europe between the north, central and south is generic since the sources are not all congruent with each other. It is then necessary to show an intermediate result of Cyprus and a very weak performance by Bulgaria (last).

### 3.2. The Impact of Age on the IoT Use in Europe

The next step (ST2) in the analysis is to understand how much the range of years affects the result. Section 2 showed that the use of a special indicator RAVG_i,j_ is not affected by the percentage of device use among countries, but is calculated specifically for each country and is not affected—Table 1.

The results of this indicator highlight that a general and common practice across Europe is the decreasing use of IoT devices as age increases. Some observations can also emerge from this analysis:the value of RAVG_i,j_ tends to be 1 in the age range 16–24 in nineteen countries. In fact, Bulgaria, Germany, Ireland and Lithuania have their highest values in the age range of 25–34 and a particular situation concerns Cyprus, the Netherlands and Sweden in the range of 35–44. Denmark and Finland also present the unitary value in that range, in addition to the 16–24 range.In the first three ranges of years considered, thus from 16 to 44, there is a reduction in RAVG_i,j,_ calculated as the average of all countries for each range of age, from 0.98 to 0.87. It is thus recorded that the decrease is not constant but becomes much more significant from the 45–54 age range with 0.68. The next two age ranges continue in the same significant decrease with average values of 0.45 and 0.24.

The result is not surprising since digitization is now a practice of use already in children and this leads to an ease in its use. Results highlight that there is a target market that is more usual to this choice, but the data provided does not allow us to define the choices of older people. A campaign to raise awareness and educate people about its use could have a propelling effect, provided the target audience feels it is necessary for the fulfilment of their needs.

### 3.3. Clustering of the European Countries

The third stage (ST3) of the analysis is to apply the MCDA method in which an equal weight is applied among all twelve criteria. The objective of this stage is twofold. The first to assess whether there are differences from what was obtained in Section 3.1, where the percentage values were reported and no normalization was carried out—Table 2. The second perspective aims to cluster the countries—Figure 2. The cluster can be proposed for each age group, however here we prefer to identify a single cluster and thus refer to the overall value.

Normalization is proposed for each criteria and then the different contributions are summed. The sum of all values determines a total intermediate value, which is then divided by the total number of criteria, as the equally weighted average is considered. The indicator used in this analysis was NAVG_i,j._

The results show variations in the ranking and thus the two methods provide different results. In fact, normalization tends to place more emphasis on the maximum and minimum values that represent the two reference values. The other intermediate values are then calculated by the linear interpolation. At the same time, normalization has the advantage of being able to consider criteria that have different units. In the percentage calculation of people using IoT items, this advantage is obviously lost.

The first result that emerges is the first position is no longer occupied by the Netherlands but by Denmark with a value of 0.808 vs. 0.706. It is noted that, compared to the European average, only one of the twenty-five countries examined changes its position from the European average: Portugal. That country coincides with the EU 27 while, in the normalized approach, it has a lower value.

Proceeding to aggregate the data by cluster, we can show that the difference between the maximum and minimum value is 0.791 and by identifying four clusters, we divide this value by four. The following clusters are identified (Figure 3):High Cluster from 0.610 to 0.808 (dark green colour).High Medium Cluster from 0.413 to 0.609 (light green colour)Low Medium Cluster from 0.215 to 0.412 (orange colour)Low Cluster from 0.017 to 0.214 (red colour).

By logic, a country belonging to a lower cluster performs worse than a higher one.

### 3.4. Clustering by Age Group

The fourth step (ST4) in the analysis is to use the ΔGap_i,j_ indicator, which measures the difference between the maximum value among all European countries for a given age group and the relative values recorded by each country. This aspect makes it possible to measure the deviation and assess how the clusters behave as a function of all age groups. A high value of ΔGap_i,j_ corresponds to a higher variance, resulting in a low performance of the specific country. Percentage values and not normalized values are used at this stage—Figure 4.

The results of this analysis not only show that the use of IoT devices is decreasing with respect to the age group, as was previously found. In fact, they show that the gap turns out to be strictly increasing as this parameter (age group) increases. In particular, it allows each country to see for which age group it decreases the most and the comparison is made by taking the best performing country as a benchmark. The differences between the clusters are significant. Within the High Cluster, there is a variation in the ΔGap indicator from 0 to 8 (with Malta showing 11 for the 65–74 age group). In contrast, within the Low Cluster, the ΔGap varies from 14 to 23. It is worth mentioning that these differences measure the percentage changes and thus represent significant values than the 0–100 range.

Finally, the ΔGap shows its most significant variations for the 45–54 age group in the different clusters (only in the High Cluster is the largest value associated with the 65–74 age group). The motivation can be found in the maximum value, which is 30–32% in the first four clusters examined, while it decreases to 23% and 18% in the last two age groups. Thus, for these age groups (55–64 and 65–74), even the leading countries are beginning to reduce the percentage of IoT device use while the same is not true for the earlier age groups—Table 3.

### 3.5. Alternative Scenario with Different Weights

In this alternative scenario (ST5), it was decided to give weights to each criterion, to obtain a weighted analysis of the importance that a given criterion possesses within the reference set.

The modus operandi that followed for assigning weight to a given category is that of identifying the number of results that the Google search engine reports for the category of devices in question. Given the topic analysed, this tool appears to be suitable.

Specifically, in order to obtain as meaningful a value as possible for the weights, it was chosen to identify, for each criteria, the sub-criteria and go for each of these to find the value of the numbers of results in the search engine (Table A1). The distribution of weights sees three criteria catalysing 80% of the total weight (Table A2):Individuals used an internet-connected home alarm system, smoke detector, security cameras, door locks or other internet-connected security/safety solutions for their home with 0.314.Individuals used a smart watch, a fitness band, connected goggles or headsets, safety-trackers, internet-connected accessories, internet-connected clothes or shoes with 0.294.Individuals used the internet on an internet-connected TV, game console, home audio system, smart speakers with 0.190.

Once the number of results for each sub-criteria was obtained, these values were added up, in order to assign a representative value for each criterion (NWAVG_i,j_). Table 4 shows the normalized values for all individuals.

The last step of the study is to consider an alternative scenario to assess whether the results obtained depend on the equally distributed weight that was assumed in the base scenario. The values show several variations. In particular, at the top the performance of Sweden (+2 places), Luxembourg (+3) and Finland (+4) improve, while Denmark (−1), the Netherlands (−3) and Spain (−3) decrease. The reason is evidently due to the weights assigned to the criteria, where it should be noted that three of them have about 80% of the total value. When there is a scenario in which the criteria are distributed in this way, the result tends to vary and reward those countries with a greater use of IoT devices in those criteria.

A cluster analysis could be created by fixing the numerical values of the base case, but alternatively it can be recalculated using the same approach. The delta between the maximum and minimum is 0.778, with a deviation value therefore of 0.195. The change in the extremes of the range are not significant, compared with the baseline case. The following clusters are identified—Figure 5:High Cluster from 0.617 to 0.811 (dark green colour).High Medium Cluster from 0.422 to 0.616 (light green colour).Low Medium Cluster from 0.228 to 0.421 (orange colour).Low Cluster from 0.033 to 0.227 (red colour).

Among the High Cluster countries, such as Luxembourg and Finland, enter, while Spain takes the reverse path. Relative to the Low Cluster, Hungary and Italy fall back; an opposite direction is followed by Cyprus. Finally, the Czech Republic and Slovakia have a value higher than the EU 27 in the alternative scenario.

## 4. Conclusions

Innovation management analysis is the antidote to conservative choices that do not consider the dynamism of eco-systems. This study arose from the need to obtain an overview regarding the use of IoT devices of different categories for European countries, in order to understand the way and speed at which the market for these devices is expanding.

### 4.1. Methodological Implications

The results of this study provide several insights. The methodological approach consists of simple indicators that, integrated with each other, nevertheless provide multiple observations. Initial observations indicate that an indicator based on the MCDA appears to be a well-established approach for constructing clusters across countries. Where the criteria have the same metrics, there is no need to replace the percentage values with normalized ones; however, we can highlight how the ranking of countries tends to change. Thus, the normalized approach strengthens those countries that represent benchmarking in the analysed sectors. Further food for thought is regarding the variation of the countries. It is worth considering that where criteria provide different values to the result, the weighted average would penalize the higher value-added criteria. The results also show variation for this alternative scenario from the baseline.

### 4.2. Managerial Implications

The speed at which Europe is traveling is not uniform. As is evident from the results shown in this article, diffusion is happening at multiple speeds: there are countries that invest in this slice of the market, which in most cases are the most economically advanced countries. Conversely, countries that do not encourage its use show very poor results. These results also show the way in which the various factors influencing this market are particularly crucial, such as the age of the end user. Specifically, from a critical point of view, we argue that the spread and use of the IoT is a turning point for any country from any point of view: economic, social, cultural, and innovative. Such devices are not only a way to make many everyday actions more agile and satisfying but also a launching point in order to achieve a strong propulsion toward technological innovation. In addition, there is a need to apply the development of IoT devices to other sectors and, in particular to the health sector, which is the beating heart of the civil society. There are opportunities for improvement, and these choices need to be calibrated, according to the sustainable impact as well.

### 4.3. European Implications: An Overview

The results highlight the positive performance of Northern European countries in the different scenarios examined. The Netherlands excels in the percentage approach, while in the normalized approach, it is Denmark that prevails if an equal weight is considered among the criteria; if a different weight is considered, it is Sweden that prevails. The identification of the High Cluster sees the presence of only four countries in the two scenarios examined. In addition to the three mentioned above, there is one country from Southern Europe (Malta). The finding that the 16–24 age group is the most likely to use IoT devices is not surprising; however, two other important reflections emerge. The first is that the three Northern European countries that perform well overall but also have hyper-relevant results in the 35–44 age group, thus highlighting that systems development is not confined to the age group. However, the second reflection emerges here as for the last two age groups examined (55–64 and 65–74) the decrease in the use of IoT devices becomes significant.

### 4.4. Limitations and Future Directions

In order to continue the study, regarding the market for IoT devices among European countries, it is interesting to carry out an integration of this study with the data on the Eurostat databases “IoT—Barriers to Use”. In this way, we can understand the reasons why part of the population does not lend itself to the use of such devices. At the same time, it is useful to conduct social analyses through surveys to understand consumer behaviours and to assess any behaviours, personal characteristics associated with those who use IoTs, compared to those who do not. In addition, the willingness to pay toward these products also turns out to be useful, tying it to sustainability characteristics as well.

In addition, in order to highlight how much confidence with the Tech domain influences the choice to buy and use an IoT device, we aim to conduct further integration with data on other types of individuals (e.g., ICT Professionals/non-ICT Professionals). In addition, indicators will need to be developed to address the issue of environmental health, which is propaedeutic, in light of the pandemic period and climate change.

### 4.5. Policy Implications

This study has not evaluated the potential policies whose impacts could lead to an increase in the use of IoT devices. First, as noted in the literature, digitization can impact positively but also negatively on sustainability. Therefore, a careful analysis is needed in which public subsidy should be given only for those devices that result in a benefit to the community. The digital market evolves every day and is visible in this study where the search engine Google proposes the relevance of different criteria, than those proposed by Eurostat. Certainly, the time factor affects these results. In addition, the volatility of product prices (particularly in this historical period when European supply chains seem short and lacking in critical raw materials) and changes in consumer choices should also be emphasized. In this regard, therefore, priority should be given to those products that provide benefit and do not represent excessive use of natural resources to be obtained. The last aspect to consider is the change in people’s lifestyles. The pandemic has favoured smart working patterns, and this pushes the context of a home to be digital, in order to optimize time. Adopting the IoT requires training programs for households, and local governments should plan citizen awareness campaigns.

## Figures and Tables

**Figure 1 ijerph-19-14343-f001:**
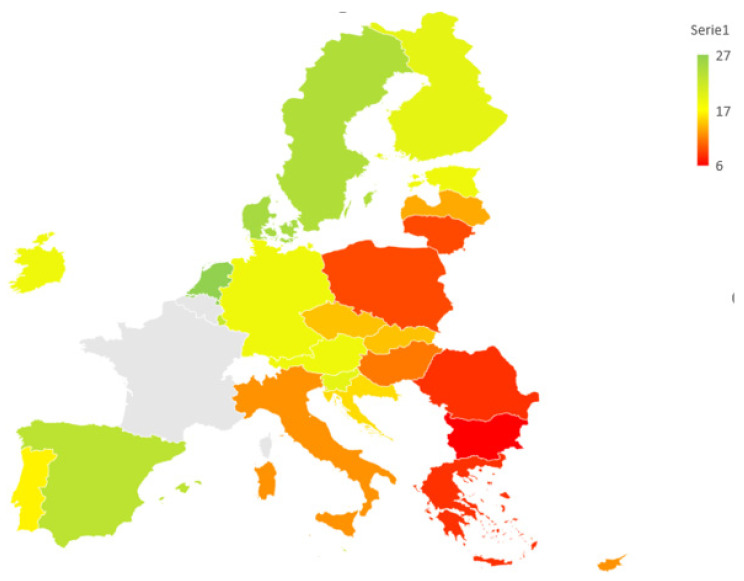
A map of the IoT users in Europe (all individuals)—scenario equals weights. Data in percentage.

**Figure 2 ijerph-19-14343-f002:**
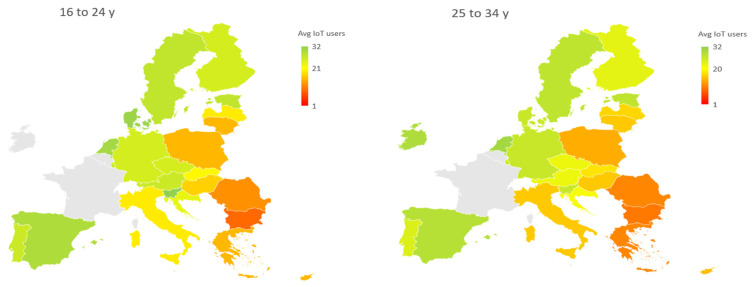
A map of the IoT users in Europe (for age group)—scenario equals weights. Data in percentage.

**Figure 3 ijerph-19-14343-f003:**
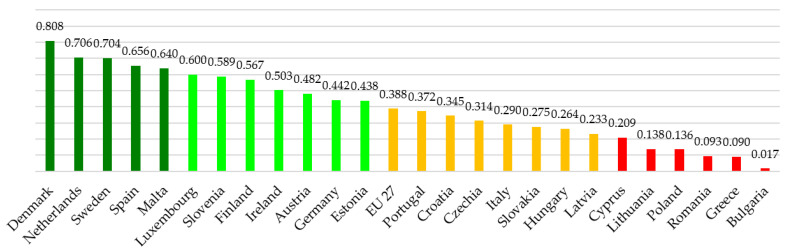
Clustering of European countries—equal weights.

**Figure 4 ijerph-19-14343-f004:**
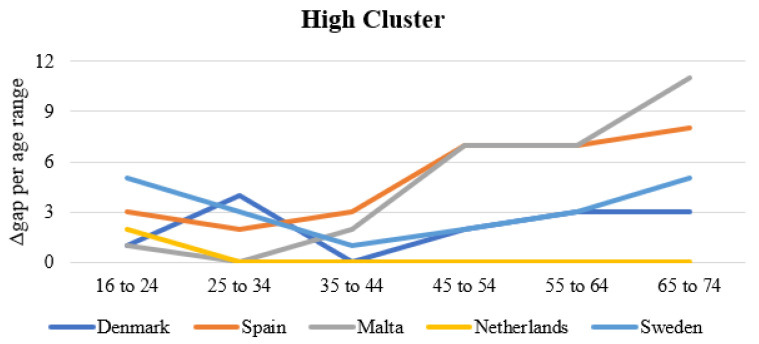
Clustering of the European countries in function of the age group.

**Figure 5 ijerph-19-14343-f005:**
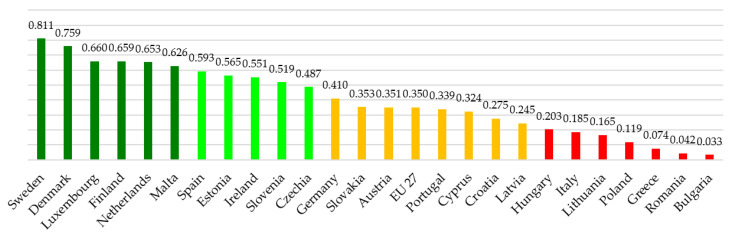
Clustering of the European countries—different weights.

**Table 1 ijerph-19-14343-t001:** The value of the average ratio indicator in Europe.

Age	16 to 24	25 to 34	35 to 44	45 to 54	55 to 64	65 to 74
Bulgaria	0.90	1.00	0.90	0.70	0.40	0.10
Czech Republic	1.00	0.88	0.68	0.52	0.32	0.12
Denmark	1.00	0.84	1.00	0.90	0.65	0.48
Germany	0.96	1.00	0.88	0.65	0.42	0.27
Estonia	1.00	1.00	0.81	0.59	0.30	0.15
Ireland	missing	1.00	0.79	0.66	0.52	0.31
Greece	1.00	0.73	0.60	0.60	0.27	0.13
Spain	1.00	0.97	0.97	0.79	0.55	0.34
Croatia	1.00	0.91	0.87	0.74	0.43	0.13
Italy	1.00	0.84	0.79	0.58	0.42	0.21
Cyprus	0.94	0.94	1.00	0.75	0.56	0.25
Latvia	1.00	0.89	0.89	0.63	0.42	0.21
Lithuania	0.94	1.00	0.81	0.50	0.25	0.13
Luxembourg	1.00	1.00	1.00	0.73	0.62	0.42
Hungary	1.00	0.94	0.82	0.65	0.41	0.29
Malta	1.00	0.97	0.94	0.74	0.52	0.23
Netherlands	0.97	0.97	1.00	0.97	0.74	0.58
Austria	1.00	0.85	0.88	0.65	0.42	0.23
Poland	1.00	0.93	0.80	0.53	0.33	0.13
Portugal	1.00	0.92	0.88	0.58	0.35	0.15
Romania	1.00	0.92	0.83	0.58	0.33	0.08
Slovenia	1.00	0.81	0.78	0.59	0.34	0.13
Slovakia	1.00	0.90	0.90	0.65	0.35	0.15
Finland	1.00	0.92	1.00	0.88	0.64	0.36
Sweden	0.90	0.90	1.00	0.93	0.67	0.43

**Table 2 ijerph-19-14343-t002:** A comparison between the percentage value and the normalized value—equal weights.

Normalized Approach	Value (NAVG_i,j_)	Percentage Approach	Value (AVG_i,j_)
Denmark	0.808	Netherlands	27
Netherlands	0.706	Denmark	25
Sweden	0.704	Sweden	24
Spain	0.656	Spain	23
Malta	0.640	Malta	23
Luxembourg	0.600	Luxembourg	22
Slovenia	0.589	Slovenia	19
Finland	0.567	Finland	19
Ireland	0.503	Germany	18
Austria	0.482	Estonia	18
Germany	0.442	Ireland	18
Estonia	0.438	Austria	18
EU 27	0.388	Portugal	16
Portugal	0.372	EU 27	16
Croatia	0.345	Croatia	15
Czech Republic	0.314	Czech Republic	14
Italy	0.290	Slovakia	14
Slovakia	0.275	Latvia	13
Hungary	0.264	Italy	12
Latvia	0.233	Cyprus	12
Cyprus	0.209	Hungary	11
Lithuania	0.138	Lithuania	9
Poland	0.136	Poland	9
Romania	0.093	Greece	8
Greece	0.090	Romania	8
Bulgaria	0.017	Bulgaria	6

**Table 3 ijerph-19-14343-t003:** Average value of the ΔGap indicator.

	16 to 24	25 to 34	35 to 44	45 to 54	55 to 64	65 to 74
High Cluster	2.4	1.8	1.2	3.6	4	5.4
Medium-High Cluster	4.75	4.8	6.6	10.8	9.2	10.2
Medium-Low Cluster	11.8	11.8	13.6	17.6	15.2	14
Low Cluster	17.6	16.6	19	21.2	17.8	15.8

**Table 4 ijerph-19-14343-t004:** Delta value between different and equal weights with a normalized approach.

Normalized Approach	Value (NWAVG_i,j_)	Delta (NWAVG_i,j_–NAVG_i,j_)
Sweden	0.811	0.107
Denmark	0.759	−0.049
Luxembourg	0.660	0.060
Finland	0.659	0.092
Netherlands	0.653	−0.054
Malta	0.626	−0.014
Spain	0.593	−0.063
Estonia	0.565	0.127
Ireland	0.551	0.048
Slovenia	0.519	−0.070
Czech Republic	0.487	0.174
Germany	0.410	−0.031
Slovakia	0.353	0.078
Austria	0.351	−0.131
EU 27	0.350	−0.038
Portugal	0.339	−0.033
Cyprus	0.324	0.115
Croatia	0.275	−0.069
Latvia	0.245	0.012
Hungary	0.203	−0.061
Italy	0.185	−0.106
Lithuania	0.165	0.027
Poland	0.119	−0.017
Greece	0.074	−0.016
Romania	0.042	−0.051
Bulgaria	0.033	0.015

## Data Availability

Not applicable.

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
