# Peer review of "European IoT Use in Homes: Opportunity or Threat to Households?"

_ijerph, 2022, doi:10.3390/ijerph192114343_

Round 1

Reviewer 1 Report

The article could improve the objective by allowing a greater number of indicators and a detailing of how the IoT are being used in the surveyed countries, as well as other databases could be used to complete the indicators. The authors could elaborate more on how the google search engine was chosen to assign the weight.

Author Response

Reviewer #1 comment

Author’ Response

The article could improve the objective by allowing a greater number of indicators and a detailing of how the IoT are being used in the surveyed countries, as well as other databases could be used to complete the indicators. 

We have tried to explain better our choice. We have used it according to literature.

We have considered all the items available in Eurostat and it is appropriate to use a single database in order to maintain the homogeneity of the data considered. Subsequent analyses could expand this dataset but paying as much attention to not repurposing redundant data or data collected by statistically different methods that could compromise the available dataset. This approach is justified by literature [47].

The authors could elaborate more on how the google search engine was chosen to assign the weight.

Thanks for this aspect. We have clarified it according to literature

In fact, literature analysis [47] suggests this approach suitable then particularly for digital components, in which the component to be considered is entered as keywords.

Reviewer 2 Report

The article is very interesting and scientifically valuable. It raises a very important issue concerning the development of innovation (IoT) in households and their impact on the quality of life in 26 out of 27 EU countries. I have some doubts about the title of the text. They concern the phrase "civil society", which I propose to delete or replace with a more general wording. Civil society has specific features, and not all of the surveyed countries fall into this category. I propose a second solution, namely referring to this concept in the text and indicating the criteria of civil society.

I appreciate the precise and detailed description of the research process, as well as the justification for the selection of the method, research area and identification of variables. I also like the way the research results are presented and the synthetic interpretation of them in relation to the adopted variables. For a clearer picture of the research results presented in the charts, I suggest adding numerical values (mainly in the charts - Figures 3 and 5).

I also appreciate the conclusions that correspond to the research results. I have one methodological remark, namely that I suggest introducing a research problem and a possible hypothesis in the introduction to the methodology and will relate the conclusions of the research to them. Literature and sources are up-to-date and sufficient.

Author Response

Reviewer #2 comment

Author’ Response

The article is very interesting and scientifically valuable. It raises a very important issue concerning the development of innovation (IoT) in households and their impact on the quality of life in 26 out of 27 EU countries. 

Thanks

I have some doubts about the title of the text. They concern the phrase "civil society", which I propose to delete or replace with a more general wording. Civil society has specific features, and not all of the surveyed countries fall into this category. I propose a second solution, namely referring to this concept in the text and indicating the criteria of civil society.

We are totally agree. The term “civil society” was not suitable. We have used the term “citizens” because all data provided by Eurostat regarding them.

New title is “European IoT use in homes: opportunity or threat to households?”

For this scope, we have preferred to not insert concepts of civil society able to add not relevant parts.

However, issues related to environmental health need to be evaluated to identify a proper mix between the needs of civil society and those of ecosystems [36]. The approach encompassing sustainable transition changes are also reflected in digital ones relative to residential [37] or industrial applications [38], but also cross-cutting [39].

I appreciate the precise and detailed description of the research process, as well as the justification for the selection of the method, research area and identification of variables. I also like the way the research results are presented and the synthetic interpretation of them in relation to the adopted variables. 

Thanks

For a clearer picture of the research results presented in the charts, I suggest adding numerical values (mainly in the charts - Figures 3 and 5).

Done. We had not made that choice as a matter of space, but such figures were extended without compromising image quality.

I also appreciate the conclusions that correspond to the research results. 

Thanks

I have one methodological remark, namely that I suggest introducing a research problem and a possible hypothesis in the introduction to the methodology and will relate the conclusions of the research to them. 

We are totally agree with this comment able to give an objective picture of the work.

The research question (RQ) that this work aims to investigate is the use of IoT devices in European households. In order to achieve this goal, it is used a mix of (ST) Statistical Tools that ensure accuracy and reliability in the results.

In both methodology and results, we have introduced ST1, ST2, ST3, ST4 and ST5.

Literature and sources are up-to-date and sufficient.

Thanks

Round 2

Reviewer 1 Report

The authors responded and completed the text with the requests from the previous round, improved the article by adding new references in the bibliography, and corrected some errors in the text.